# Expression of Matrix Metalloproteinases in the Circulating Immune Cells in Children with *Helicobacter pylori* Infection—Correlation with Clinical Factors

**DOI:** 10.3390/ijms242115660

**Published:** 2023-10-27

**Authors:** Anna Helmin-Basa, Izabela Kubiszewska, Małgorzata Wiese-Szadkowska, Edyta Strzyżewska, Aleksandra Skalska-Bugała, Sara Balcerowska, Marta Rasmus, Daria Balcerczyk, Marta Pokrywczyńska, Jacek Michałkiewicz, Aneta Krogulska, Adam Główczewski, Anna Szaflarska-Popławska

**Affiliations:** 1Department of Immunology, Faculty of Pharmacy, Collegium Medicum in Bydgoszcz, Nicolaus Copernicus University, 87-100 Torun, Poland; a.helminbasa@gmail.com (A.H.-B.); izakubiszewska@gmail.com (I.K.); malgorzatawiese@gmail.com (M.W.-S.); edytaroszak96@gmail.com (E.S.); aleksandraaskalska@gmail.com (A.S.-B.); sara.balcerowska@onet.pl (S.B.); 2Department of Regenerative Medicine Cell and Tissue Bank, Faculty of Medicine, Collegium Medicum in Bydgoszcz, Nicolaus Copernicus University, 87-100 Torun, Poland; marta.rasmus@gmail.com (M.R.); daria.balcerczyk@gmail.com (D.B.); marta.pokrywczynska@interia.pl (M.P.); 3Department of Microbiology and Clinical Immunology, The Children’s Memorial Health Institute, 04-730 Warsaw, Poland; jjmichalkiewicz@wp.pl; 4Department of Pediatrics, Allergology and Gastroenterology, Faculty of Medicine, Collegium Medicum in Bydgoszcz, Nicolaus Copernicus University, 87-100 Torun, Poland; aneta.krogulska@cm.umk.pl (A.K.); a.glowczewski@cm.umk.pl (A.G.); 5Department of Pediatric Endoscopy and Gastrointestinal Function Testing, Faculty of Medicine, Ludwik Rydygier Collegium Medicum in Bydgoszcz, Nicolaus Copernicus University in Torun, ul. Jagiellonska 13-15, 85-067 Bydgoszcz, Poland

**Keywords:** matrix metalloproteinases, tissue inhibitor of metalloproteinases, *Helicobacter pylori*, eradication therapy, children

## Abstract

*H. pylori* gastritis is strongly associated with the upregulation of the expression of several matrix metalloproteinases (MMPs) in the gastric mucosa. However, the role of MMP-2 and MMP-9, and their inhibitors (tissue inhibitors of metalloproteinases -TIMPs) produced by immune cells in infected children have not been clearly defined. Moreover, the effects of *H. pylori* eradication therapy on MMPs and TIMPs production has not been evaluated. A total of 84 children were studied: 24—with newly diagnosed *H. pylori* gastritis, 25—after *H. pylori* eradication therapy (17 of them after successful therapy), 24—with *H. pylori*-negative gastritis, and 11—controls. Plasma levels of MMP-2, MMP-9, TIMP-1, and TIMP-2 by ELISA; MMPs and TIMPs expression in lymphocytes; neutrophils and monocytes in peripheral blood by multiparameter flow cytometry; and mucosal mRNA expression levels of MMPs and TIMP-1 in gastric biopsies by RT-PCR were evaluated. Children with *H. pylori*-related gastritis showed the following: (1) increased MMP-2 and TIMP-2 plasma levels, (2) increased intracellular expression of MMP-2 in the circulating lymphocytes and neutrophils, (3) low frequencies of circulating TIMP-1+ and TIMP-2+ leukocytes, and (4) high expression of mRNA for MMP-9 along with low expression of mRNA for MMP-2 in the gastric mucosa. Unsuccessful *H. pylori* eradication was associated with the following: (1) high plasma levels of MMP-9 and TIMP-1, (2) increased pool of TIMP-1+ lymphocytes as well as high expression of MMP-9 in circulating lymphocytes, and (3) high expression of mRNA for MMP-9 in the gastric mucosa. Our data suggest that MMPs are important contributors to stomach remodelling in children with *H. pylori*-related gastritis. Unsuccessful *H. pylori* eradication is associated with increased MMP-9 in plasma, circulating lymphocytes, and gastric mucosa.

## 1. Introduction

*Helicobacter pylori* (*H. pylori*) is a spiral-shaped, Gram-negative, microaerophylic bacterium found in the stomach. Stomach colonisation is usually life-long and inevitably causes chronic mucosal inflammation that can remain silent or evolve into more severe diseases, such as atrophic gastritis, intestinal metaplasia, peptic ulcer, lymphoma of the mucosa-associated lymphoid tissue (MALT), or gastric adenocarcinoma [1]. Children differ from adults with respect to *H. pylori* infection in terms of the rare presence of gastric mucosal atrophy and intestinal metaplasia and the near absence of gastric malignancies. Unlike in gastric cancer, *H. pylori*-related extra-gastrointestinal manifestations, including iron deficiency/iron deficiency anaemia and chronic immune thrombocytopenic purpura, occur more commonly in children and adolescents than in adults [2]. The current evidence also supports the hypothesis that *H. pylori* infection has an adverse impact on children’s growth outcomes, mainly on height-for-age scores [3].

Over the past decades, the prevalence of *H. pylori* in the developed world has steadily decreased. However, the infection is still highly common in children and adolescents globally. Most recent meta-analysis have shown that the overall global prevalence *of H. pylori* infection is 32.3% [4]. It is well documented that the presence of *H. pylori* is significantly higher in low-income and middle-income countries than in high-income countries (43.2% versus 21.7%) and in older children than in younger children (41.6% in 13–18-year-olds versus 33.9% in 7–12-year-olds versus 26% in 0–6-year-olds) [4]. Low socioeconomic status; room sharing; no access to a sewage system; drinking non-boiled or non-treated water; having a mother, a sibling, or siblings infected with *H. pylori*; and older age represent the most important risk factors for paediatric *H. pylori* [5].

The clinical manifestation of *H. pylori* infection in paediatric populations is not very clearly defined. About 85% of children infected with the bacteria the remain life-long asymptomatic. There is no positive association between *H. pylori* infection and paediatric gastrointestinal symptoms such as vomiting, diarrhoea, flatulence, chronic functional abdominal pain, halitosis, regurgitation, constipation, and nausea. The evidence for an association between epigastric pain and infection has been conflicting [6]. Children with recurrent abdominal pain without any alarm symptoms independent of *H. pylori* status most likely have functional abdominal pain, which is one of the most commonly encountered disorders in childhood, affecting up to 25% of children worldwide. Testing for *H. pylori* infection in this group of patients is not recommended [7].

The virulence factors of *H. pylori* such as cag pathogenicity island (cagPAI), cagPAI-encoded bacterial type IV secretion (T4SS), and heptose metabolites of the lipopolysaccharide (LPS) biosynthesis can activate nuclear factor kappa B (NFκB) signalling pathway which induces the pathogenicity processes [8,9,10,11]. The inflammation begins and involves many chemoattractants, pro-inflammatory cytokines including IL-1β, IL-8, TNF, some matrix metalloproteinases (MMPs), and tissue inhibitors of metalloproteinases (TIMPs). In consequence, the massive recruitment of leucocytes (e.g., neutrophils, macrophages, lymphocytes) begins. MMPs have an impact on inflammatory microenvironments and architectural changes in the mucosal tissue infected by *H. pylori*. Studies by Kayoma S. et al. have shown that MMP-2, -7, and -9, as well as membrane type (MT)1-MMP and TIMP-2 and -4, were upregulated in gastric mucosa biopsies from *H. pylori* infected adult patients with gastritis [12]. Additionally, the systemic level of these parameters was also evaluated. In the serum of *H. pylori*–infected children, TIMP-1, but not MMP-2 -7 -8 and -9, levels were increased in comparison to the uninfected group [13]. In contrary to studies among children, adults with *H. pylori*–related gastritis have shown increased levels of MMP-8 and -9, an unchanged level of MMP-7, and decreased levels of MMP-2 and TIMP-1 [14]. Interestingly, the expression of MMPs and TIMPs specifically in a population of leucocytes of peripheral blood in *H. pylori*-infected children have not been clearly defined yet. Therefore, in this study we examined the expression of chosen MMPs and TIMPs in peripheral blood lymphocytes, neutrophils, and monocytes. Additionally, the effect of *H. pylori* eradication therapy on MMPs and TIMPs production was evaluated.

## 2. Results

The study was conducted in 84 children: 24 patients with newly diagnosed *H. pylori*-related gastritis (*Hp+* children, median age 11.5 years, range 4–17 years); 25 patients after *H. pylori* eradication therapy, including 17 after successful treatment (children after SE median age 16 years, range 12–17 years); 8 patients after unsuccessful eradication (children after NE, median age 11.5 years, range 6–17 years); 24 *H. pylori*-negative patients without gastritis (*Hp-* children, median age 12.5 years, range 3–17 years); and 11 controls (median age 15 years, range 5–17 years).

### 2.1. Plasma Levels of MMPs and TIMP in Hp+, Hp- and Control Children

Plasma from *Hp+* children showed a higher level of TIMP-2 than plasma of *Hp-* children (*p* = 0.06) alongside an increased level of MMP-2 than in both the plasma of *Hp*- children (*p* = 0.07) and the plasma of control children (*p* = 0.04). There were no differences between the groups regarding MMP-9 and TIMP-1 plasma levels (*p* > 0.05) (Figure 1).

### 2.2. MMPs and TIMPs Expression in Circulating Lymphocytes, Monocytes, and Neutrophils from Hp+, Hp-, and Control Children

*Hp+* children had lower percentage of circulating TIMP-1+ and TIMP-2+ lymphocytes, TIMP-1+ and TIMP-2+ monocytes, as well as TIMP-2+ neutrophils as compared to *Hp-* children (*p* = 0.06 and *p* = 0.04, *p* = 0.06 and *p* = 0.01, *p* = 0.04 and *p* =0.02, *p* = 0.04 vs. control group, respectively) (Figure 2, Figure 3 and Figure 4).

*Hp+* children had slightly higher intracellular expression of MMP-2 in the circulating lymphocytes (*p* = 0.07) and neutrophils (*p* = 0.07) than *Hp-* children (Figure 5 and Figure 6). However, the intracellular expression of MMP-9 in circulating neutrophils was slightly reduced in this group (*p* = 0.05) (Figure 6). There was no difference between *Hp+* children, *Hp-* children, and the control group regarding the intracellular expression of MMPs and TIMPs in circulating monocytes (*p* > 0.05).

### 2.3. Evaluation of mRNA for MMPs and TIMP-1 in the Gastric Mucosa in Hp-+, Hp- and Control Children

Our data show that gastric mucosa from *Hp+* children presented higher expression of mRNA for MMP-9 (*p* < 0.01) than control group or *Hp-* children (*p* < 0.01). However, the expression of mRNA for MMP-2 was reduced in this group of children as compared to the control group (*p* = 0.06) (Figure 7).

### 2.4. Effect of H. pylori Eradication Therapy on Plasma Levels of MMPs and TIMPs

The plasma of children after unsuccessful *H. pylori* eradication showed higher levels of MMP-9 (*p* = 0.003) and TIMP-1 (*p* = 0.06) as compared to children after successful *H. pylori* eradication. However, we observed no difference in plasma levels of MMP-2 and TIMP-2 between children with unsuccessful and successful *H. pylori* eradication therapies (*p* > 0.05) (Figure 8).

### 2.5. Effect of H. pylori Eradication Therapy on the Intracellular Expression of MMPs and TIMPs in Circulating Lymphocytes, Monocytes and Neutrophils

Children with unsuccessful eradication therapy showed more reduced expression of MMP-2 in circulating lymphocytes than children after successful eradication therapy (*p* = 0.03) Additionally, we observed a tendency toward an increased proportion of TIMP-1-positive lymphocytes and high intracellular expression of MMP-9 in these cells in peripheral blood (*p* = 0.06 and *p* = 0.09) (Figure 9 and Figure 10).

### 2.6. Effect of H. pylori Eradication Therapy on the Expression of mRNA for MMPs and TIMP-1 in the Gastric Mucosa

The gastric mucosa of children after unsuccessful *H. pylori* eradication showed higher expression of mRNA for MMP-9 (*p* = 0.004) as compared to children after successful *H. pylori* eradication. However, no differences in expression of mRNA for MMP-1 and TIMP-1 were observed between children with unsuccessful and successful *H. pylori* eradication therapies (*p* > 0.05) (Figure 11).

### 2.7. The Association of MMPs and TIMPs with the Inflammation, Activity and Density of H. pylori in Gastric Mucosa

Correlation analysis demonstrates how MMPs and TIMPs are associated with the gastric inflammation (density of mononuclear cells), activity (density of polymorphonuclear cells), and density of *H. pylori* in children with gastritis. Our data show that the density of mononuclear cells) and *H. pylori* in gastric mucosa of children had a negative correlation with the percentages of peripheral TIMP-2+ lymphocytes (r = −0.41 and r = −0.36, *p* < 0.05), TIMP-2+ monocytes (r = −0.41 and r = −0.44, *p* < 0.05), and TIMP-2+ neutrophils (r = −0.57 and r = −0.58, *p* < 0.05). Moreover, the density of mononuclear cells and *H. pylori* also had a negative correlation with the intracellular expression of MMP-9 in peripheral neutrophils (r = −0.40 and r = −0.39, *p* < 0.05). The mRNA for MMP-9 was significantly associated with the density of mononuclear cells (r = 0.74, *p* < 0.05), density of polymorphonuclear cells (r = 0.80, *p* < 0.05), and density of *H. pylori* (r = 0.75. *p* < 0.05) in the gastric mucosa (Table 1).

## 3. Discussion

To the best of our knowledge this is the first study that demonstrates the alteration of plasma levels of some matrix metalloproteinases (MMPs) and tissue inhibitors of metalloproteinases (TIMPs) and their intracellular expression in peripheral leukocyte population in children with newly recognized *H. pylori*-related gastritis as well as after successful and unsuccessful *H. pylori* eradication therapies.

The present study was based on the hypothesis that *H. pylori* infection in children may modulate the production of selected MMPs and TIMPs in some leukocyte populations which, in turn, may lead to facilitation of the interaction between the bacteria and epithelium. The main findings of the study were that *H. pylori*-infected children with gastritis showed the following: (1) increased level of plasma MMP-2 and TIMP-2, (2) low frequencies of circulating TIMP-1+ and TIMP-2+ leukocytes, (3) increased intracellular expression of MMP-2 in the circulating lymphocytes and neutrophils, and (4) high expression of mRNA for MMP-9 along with low expression of mRNA for MMP-2 in the gastric mucosa. Moreover, unsuccessful *H. pylori* eradication was associated with high plasma levels of MMP-9 and TIMP-1, increased pool of TIMP-1+ lymphocytes as well as high expression of MMP-9 in circulating lymphocytes, and low expression of mRNA for MMP-9 in the gastric mucosa.

Our first observation was that children with *H. pylori*-related gastritis showed increased levels of plasma MMP-2 and TIMP-2 in comparison to uninfected children with gastritis.

MMP-2, also known as gelatinase A, has proteolytic activity towards classical extracellular matrix (ECM) molecules, e.g., collagens I and IV [15]. This results in the migration of lymphocytes and other leukocytes which enter and leave the blood circulation. MMP-2 is produced as secreted MMP-2 precursor (pro-MMP-2) that is processed by activated membrane type MMPs (MT-MMPs) [16]. Secreted MMP-2 can exert intra- and extracellular function, for example, it downregulates the activity of NF-κB inhibitor IκBα in response to oxidative stress associated with bacterial infection [17,18]. TIMP-2 is one of four tissue inhibitors of metalloproteinases (TIMPs) that regulate MMP-2 activity by binding to the carboxy-terminal domain of pro-MMP-2 and linking pro-MMP-2 with MT1-MMP [16,17]. Thus, concomitantly increased plasma levels of MMP-2 and TIMP-2 seem to be more relevant for *H. pylori*-related gastric inflammation in children. Additionally, high systemic levels of TIMP-2 suggests that *H. pylori* regulates MMP-2 activity via biding it to TIMP-2.

There is a lack of studies evaluating MMPs and TIMPs in the plasma of *H. pylori*-infected children. However, increased TIMP-1 but not MMP-2 and -9 levels in serum from *H. pylori*-infected children, in comparison to an uninfected group without gastritis, were detected by Rautelin et al. [13]. This discrepancy between our study and Rautelin et al.’s study may be due to the different types of study material and control group used (we used plasma and uninfected controls with gastritis, whereas they focused on serum and uninfected controls with normal gastric mucosa). An investigation comparing serum levels of MMPs (including MMP-9) between *H. pylori*-positive and *H. pylori*-negative adults with gastritis showed no difference [19]. Furthermore, adults with *H. pylori*-associated gastritis had increased levels of MMP-9 and decreased levels of MMP-2 and TIMP-1 in serum in comparison to healthy volunteers [14]. Thus, the level of plasma MMPs and TIMPs depends on the status of *H. pylori* and the age of the people in the group with gastritis.

Due to the altered plasma levels of MMPs and TIMPs, we decided to analyse this difference in the intracellular levels of circulating lymphocytes, monocytes, and neutrophils. Our data confirm that the main immune cellular source of MMP-2 in the peripheral blood of *H. pylori*-infected children with gastritis were lymphocytes and neutrophils, which demonstrated increased MMP-2 intracellular expression. However, the frequencies of circulating TIMP-1 and -2 expressing leukocytes were reduced in this gastritis group of children.

Many immune cell types secrete MMPs and TIMPs. T and B lymphocytes constitutively produce small amounts of MMP-9, which is upregulated upon chemokines and cytokines stimulation [20]. However, they do not exhibit baseline production of MMP-2—it is produced only after long period of time of stimulation (e.g., by IL-2) [21]. Neutrophils are known to be the main producers of MMPs and cytokines [22]. These cells begin inflammation response and fibrosis in gastric mucosa via continuous pathogen-associated molecular patterns (PAMPs) stimulation. It is known that some new PAMPs of *H. pylori* such as intermediates of *H. pylori* LPS biosynthesis, like heptose 1,7-bisphosphate (HBP) and its derivative ADP-heptose activate NF-κB. They regulate the transcription of genes for chosen MMPs [23,24]. MMPs from different cells activate one another, participate in architectural changes, regulate cell adhesion, and shed proteins in cellular junctions—this disturbs the cellular contacts and restructures actin [25]. MMPs can also activate cytokines and grow factors in the ECM. Thus, *H. pylori* via MMPs leads to modifications in the ECM and change adhesion-regulated cellular signals. In this way *H. pylori* contributes to chronic inflammation by promoting mucosal damage and facilitating the interaction of the epithelium with bacteria and immune cells [25].

TIMP-1 and -2 are expressed in peripheral neutrophils and monocytes as well as in low levels in peripheral blood T cells [26]. They are endogenous inhibitors of both MMP-2 and MMP-9 but TIMP-2 is more effective in inhibiting MMP-9. In addition to the classical function of MMP inhibition, TIMPs also affect many other cellular functions, such as growth, proliferation, migration, and apoptosis [27]. The reduced frequencies of TIMP-positive leukocytes in blood may be suggestive of the fact that during *H.pylori* infection in children these cells are transported from blood to the *H. pylori*-infected gastric mucosa, where we detected the high MMP-9 expression. In gastric mucosa TIMP-1 may mediate gastroprotection by rescuing cells from toxic effects of MMP-9 or by inducing gastric remodelling with enhanced collagen generation [28].

Our study also assessed mRNA for MMPs and TIMPs and the results revealed a higher expression of mRNA for MMP-9 along with a low expression of mRNA for MMP-2 in the gastric mucosa of children with *H. pylori*-related gastritis. This observation is partially consistent with the results obtained by Kazachkov et al., who have observed increased MMP-9, MMP-2, and TIMP-1 in the fundal gastric mucosa of Russian children with chronic *H. pylori*-associated gastritis [29]. One study in adults with *H. pylori*-associated gastritis showed upregulated MMP-2, -9, and TIMP-2 on the surface of infiltrative mucosal lymphocytes [12]. Other studies have demonstrated increased expression of MMP-9 and MMP-2, but no differences in TIMP-1 and -2, in the biopsies of *H. pylori-*infected subjects in comparison to uninfected ones [30,31]. Furthermore, gastric mucosa biopsies from *H. pylori*-positive patients with gastritis after incubation with culture medium also produced higher levels of MMP-9 and TIMP-1 than biopsies from *H. pylori*-negative patients [32]. Our and others’ research suggest that *H. pylori*-infected children with gastritis have different expression profile of MMPs and TIMPs in gastric mucosa than adults. Our research also found positive correlation between density of *H. pylori* in gastric mucosa and plasma levels of MMP-9 and MMP-9 mRNA in gastric mucosa. Positive correlation between MMP-9 and the activity of inflammation was also observed in children with *H. pylori*-related chronic gastritis [30]. High expression and activation of MMP-9 in the gastric mucosa may, on the one hand, stimulate the ulcerogenic properties of this MMP; one the other hand, it can facilitate bacterial immune evasion. 

In our study, MMPs and TIMPs in plasma, peripheral leukocytes, and gastric mucosa were investigated in children that are sensitive or resistant to *H. pylori* treatment for the first time. We demonstrated that unsuccessful *H. pylori* eradication was associated with the following: (1) high plasma levels of MMP-9 and TIMP-1, (2) increased pool of TIMP-1+ lymphocytes and high expression of MMP-9 in circulating lymphocytes, as well as (3) high expression of mRNA for MMP-9 in the gastric mucosa. In contrary to the findings reported in our study, Negersh et al. indicated a significant increase in the MMP-2 serum level in adults with refractory *H. pylori*-associated peptic ulcer disease [33]. Decreased levels of total and active MMP-9 after successful eradication treatment in the gastric mucosa in adults with *H. pylori*-associated gastritis adults was also observed by Kubben et al. [34,35]. Based on our data, increased MMP-9 and TIMP-1 in plasma as well as in circulating lymphocytes can be used as an easy marker of unsuccessful eradication therapy in children.

## 4. Materials and Methods

### 4.1. Patients

All 84 patients aged between 4 and 18 years included to the study were hospitalized in the Department of Pediatrics, Allergology of Gastroenterology, or were consulted by a paediatric gastroenterologist in Outpatient Gastroenterological Clinic for Children. All patients were recruited to the study and all endoscopies were performed by the same paediatric gastroenterologist (ASP). Patients were assigned to the studied groups at endoscopy or based on the ^13^C UBT result (in children after eradication therapy when no control endoscopy was performed).

Seventy-four of eighty-four children, including all but one with newly diagnosed *H. pylori*-related gastritis, all with *H. pylori*-negative-gastritis, and all controls, were examined with upper gastrointestinal endoscopy.

In 55 of 58 children with *H. pylori* -positive and -negative-gastritis and in controls, the main indication for upper gastrointestinal endoscopy was chronic abdominal pain and/or nausea/vomiting and at least one accompanying symptom/sign (unsatisfactory weight gain or weight loss, anaemia, abdominal pain awakening at nights, non-effective antisecretory or trimebutine treatment, positive non-invasive *H. pylori* testing, positive family history for coeliac disease, peptic ulcer disease, or *H. pylori*-related gastritis). In another 3 patients the endoscopy was performed for foreign body removal (1 patient), before bariatric surgery (1 patient), and as control procedure after non-*H. pylori*-peptic ulcer disease treatment (1 patient). There was no difference in clinical manifestations between children with *H. pylori*-positive and -negative gastritis.

In 13 of 16 children after eradication therapy in whom gastroscopy was performed, patients were scoped because of recurrence of abdominal pain. In another 3 patients the indications for endoscopy were choking (1 patient) and surveillance for peptic ulcer disease (2 patients).

The majority of upper gastrointestinal endoscopies (63 of 74) were performed under general anaesthesia with flexible Olympus gastroscopes (GIF 160, GIF-H180J, GIF-Q165). Eleven patients underwent the procedure as outpatients only with topical aesthetic. All gastroscopies were performed by one paediatric gastroenterologist (ASP) with more than 15 years experience in paediatric endoscopic procedures.

The exclusion criteria included (1) a history of antibiotics, bismuth compounds, H**_2_** antagonists, proton pump inhibitors, or immunosuppressive drugs within the last four weeks before endoscopy; (2) previous diagnosis of other inflammatory disease, such as coeliac disease and inflammatory bowel disease; (3) any evidence of current parasitic infection; 4) history of gastric surgery; and (5) history of neoplastic disease. None of the children included into the study had current signs of overt upper gastrointestinal bleeding, whereas 2 of them had a previous endoscopy because of severe upper gastrointestinal bleeding over the course of peptic ulcer disease. Each subject and/or subject’s parents provided the clinical history. If the subject fulfilled the inclusion criteria, they were invited to participate in the study.

At endoscopy at least five mucosal biopsies were taken from gastric mucosa, including four specimens for histological assessment and one for mRNA expression of MMPs and TIMPs via RT-PCR. As recommended [35] for the histopathological analysis, two gastric biopsies from the antrum and two from the corpus were obtained. Four specimens were formalin-fixed and embedded in paraffin, sectioned, and stained with haematoxylin and eosin for histological analysis, or Giemsa modified by Gray stain for *H. pylori* detection. Biopsy specimens were graded for gastritis by two independent pathologists based on the updated Sydney system. Histological variables (presence and density of mononuclear and polymorphonuclear cells, glandular mucosa atrophy, and intestinal metaplasia) were scored on a four-point scale: 0, none; 1, mild; 2,moderate; and 3, marked. The fifth gastric biopsy was located in RNAlater stabilization and storage reagent and frozen (−80 °C) for the detection of mRNA expressions of MMPs and TIMPs via RT-PCR.

A patient was considered *H. pylori* infected with both *H. pylori* and gastritis on histopathology and at least 1 other positive test such as rapid urease test or *H. pylori* stool antigen test or ^13^C-urea breath test (UBT). A patient was considered *H. pylori* negative with *H. pylori* negativity upon histopathology. The control group consisted of children with no signs of macroscopic and microscopic gastric pathology. The outcome of anti-*H. pylori* therapy was assessed at least 4 weeks after completion of therapy using ^13^C-UBT. The therapy was considered successful when ^13^C concentration in the exhaled air was ≤3.5‰, otherwise it was considered unsuccessful. ^13^C concentration was measured with an infrared radiation analyser (OLYMPUS Fanci2, Tokyo, Japan). 

*H. pylori-*infected children were treated with high-dose triple therapy (amoxicillin, metronidazole, and proton-pump inhibitor—PPI) for 10–14 days. Doses of antibiotics and PPI-s were calculated based on the bodyweight. In non-eradicated patients, antibiotic therapy was repeated with clarithromycin, amoxicillin, and PPI for 10–14 days.

During endoscopy, venous blood was also obtained from each patient for immunological testing (cytometry and ELISA).

### 4.2. Flow Cytometry Technique

During endoscopy, 3 mL of peripheral blood was obtained from each patient using a sterile TransFix/EDTA vacuum tubes (Cytomark, Buckingham, UK). The active components of TransFix stabilise leukocytes and leucocytic antigens for up to 14 days. Until cytometric tests were performed, the blood was stored at 4 °C.

A 100 μL volume of TransFix/EDTA blood each were added to 2 flow cytometry tubes (test and control tube) and mixed with monoclonal antibodies (mAbs) anti-CD45 (clone HI30) conjugated with Brilliant Violet 421 (BV421) (BD Pharmingen**^TM^**, San Diego, CA, USA) at a predetermined optimal concentration and incubated for 30 min at 4 °C. After erythrocyte lysis, by adding 2 mL of lysing solution without paraformaldehyde (Pharm Lyse, BD Bioscience, San Jose, CA, USA) and 0.5 mL of lysing liquid with paraformaldehyde (FACS Lysing Solution, BD Biosciences, San Jose, CA, USA) (5–10 min, RT, in the dark), the cells were washed with 2 mL of phosphate-buffered saline (PBS, Biomed Lublin, Poland), centrifuged (5 min, 500 g) and the supernatant was decanted.

To avoid non-specific binding of used mAbs before permeabilization the cell pellet was resuspended in 1 mL of PBS solution supplemented with 10% filtered and inactivated human serum (HS). The samples were incubated for 30 min at 4 °C in the dark, and then centrifuged and supernatant was decanted. Cells were then fixed, permeabilized using the BD Cytofix/CytopermTM Fixation/Permeabilization Kit (BD Bioscience, San Jose, CA, USA), resuspended in 100 µL Perm Wash (BD Bioscience, San Jose, CA, USA), and centrifuged. Then, cells in test tube were incubated with mAbs conjugated with distinct fluorochromes: anti-MMP-2 PE (clone 1A10), anti-MMP-9 FITC (clone 56129), anti-TIMP-1 AF647 (clone 63515) (all from R&D Systems, Minneapolis, MN, USA), and anti-TIMP-2 PerCp (clone OTI1A6) (Novus Biologicals, Littleton, CO, USA) (30 min at 4 °C in the dark).

After incubation samples were washed twice with 2 mL of Perm Wash, centrifuged (5 min, 500 g), and the supernatant was decanted. Cells for cytometric analysis were resuspended in 300 µL of PBS. All samples were analysed immediately after the staining procedure was completed. The phenotypic analyses were performed using flow cytometry (BD FACSCanto II, Becton Dickinson, San Diego, CA, USA). Ultimately, 5000 cells in the monocytic gate (Sight (SSC)^med^/CD45^high^) were collected. Data acquisition was performed using the CellQuest Pro™ Software version 6.1.3 (BD Biosciences) and the analysis with FlowJo 7.5.5 (Tree Star, Inc. Ashland, OR, USA).

In the first step, leukocytes were gated according to the size (Forward Scatter—FSC) and granularity (SSC) of cells. Damaged cells and cell clumps (doubles) were excluded from analysis. Then, the neutrophil, monocyte, and lymphocyte populations were gated based on SSC/CD45 parameters. In the leukocyte populations, the percentage of cells expressing MMP-2, MMP-9, TIMP-1, and TIMP-2 and their density in cells (MFI—median fluorescence intensity) were analysed (the gating strategy was shown in Figure 12). Isotype antibody controls were used to ensure antibody specificity in independent tubes.

### 4.3. Real-Time PCR

Total RNA was extracted from tissue samples using RNeasy Mini Kit (Qiagen, Hilden, Germany) according to the manufacturer’s protocol. The purity and quantity of isolated RNA were analysed on a NanoDrop (Thermo Scientific, Waltham, MA, USA), and the integrity was determined by an Agilent 2100 BioAnalyzer (Agilent Technologies, Santa Clara, CA, USA) with the use of an RNA 6000 Nano Kit (Agilent Technologies, USA). cDNA was synthesized from 500 ng of total RNA using an iScript cDNA Synthesis Kit (Bio-Rad, Hercules, CA, USA). The panel of 14 genes (Reference genes H96 Plate, Bio-Rad) was used to evaluate and identify the most suitable reference genes for the analysed samples. RPLP0 and RPS18 were used for normalization as the most stable reference genes. Quantitative real-time PCR was performed with the use of SsoAdvanced Universal SYBR Green Supermix (Bio-Rad, USA) and PrimePCR™ primers (Bio-Rad, USA) (Table 2). The Roche LightCycler 480 software 1.5 (Roche Diagnostics, Basel, Switzerland) was used to perform advanced relative quantification analysis.

### 4.4. ELISA

To evaluate the plasma MMP and TIMP levels, 9 mL of blood was taken in a vacutainer heparin blood collection tube. Then, the samples were centrifuged at 300× *g* for 10 min to separate plasma. After the plasma was transferred into new conical centrifuge tubes the samples were centrifuged again (1000× *g*, 10 min) to remove any cellular debris. The samples were then transferred to a refrigerator and stored at −80 °C until use. The DuoSet ELISA kits (R&D Systems; Minneapolis, MN, USA) were used to determine the total concentration of MMP-2, MMP-9, TIMP-1, as well as TIMP-2 according to the manufacturer’s instructions. For MMP-9, TIMP-1, and TIMP-2 the plasma was diluted 1:100 and the dilution was included in final results.

### 4.5. Statistical Analyses

Statistical analysis was performed using PS IMAGO PRO v. 9.0 (Predictive Solutions Sp z o.o., Cracow, Poland) based on the IBM SPSS Statistics v29 analytical engine. To test the hypotheses of normal distribution, variables were analysed using the Shapiro–Wilk test. Variables with non-Gaussian distribution are described as medians with quartiles and nonparametric tests were applied. To test the hypotheses of equal distributions in *Hp+* gastritis, *Hp****-*** gastritis, and control groups, the non-parametric Kruskal–Wallis test followed by Dunn’s multiple statistic was applied. The hypotheses of equal distributions in SE and NE groups were tested according to Mann–Whitney’s statistic. Dependencies between age, gastric inflammation, and other examined parameters were analysed by Spearman’s rank correlation. A significance level equal to 0.05 was used.

### 4.6. Ethical Considerations

All patients/guardians provided informed written consent for participation in the study, as regulated by the local law. The protocol of the study was approved by the Ethics Committee of Collegium Medicum in Bydgoszcz, Nicolaus Copernicus University in Torun and registered under the number KB 362/2021.

## 5. Conclusions

Our data suggest that (1) MMPs are important contributors to stomach remodelling in children with *H. pylori*-related gastritis, (2) remodelling of stomach ECM is regulated by lymphocytes and neutrophils, through the production of more protective MMP-2, and (3) unsuccessful *H. pylori*-eradication is associated with increased MMP-9 in plasma, circulating lymphocytes, and in gastric mucosa. The increased synthesis of MMP-9 can stimulate ulcerogenic properties of this MMP in the gastric mucosa so the evaluation of this MMP in plasma, as well as in circulating lymphocytes, can be used as an easy refractory *H. pylori*-induced gastritis or peptic ulcer marker in children.

## Figures and Tables

**Figure 1 ijms-24-15660-f001:**
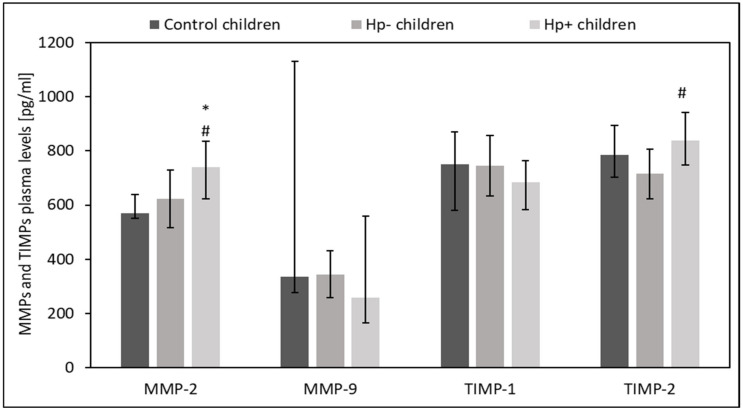
Evaluation of MMPs and TIMPs plasma levels Hp+, Hp-, and control children. Hp- children: uninfected children with gastritis, Hp+ children: *H. pylori*-infected children with gastritis. Significant differences (Kruskal–Wallis test, *p* < 0.05): * vs. control children. Trend toward significance (Kruskal–Wallis test, 0.1 > *p* > 0.05): # vs. Hp- children. Variables in compared subgroups have non-Gaussian distribution, thus they are described as medians with quartiles.

**Figure 2 ijms-24-15660-f002:**
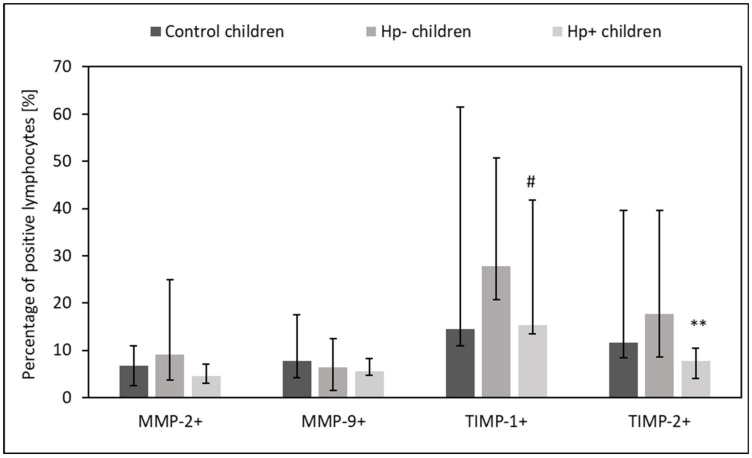
The percentage of circulating MMPs- and TIMPs- positive lymphocytes in Hp+, Hp-, and control groups. Hp- children: uninfected children with gastritis, Hp+ children: *H. pylori*-infected children with gastritis. Significant differences (Kruskal–Wallis test, *p* < 0.05): ** vs. Hp- children. Trend toward significance (Kruskal–Wallis test, 0.1 > *p* > 0.05): # vs. Hp- children. Variables in compared subgroups have non-Gaussian distribution, thus they are described as medians with quartiles.

**Figure 3 ijms-24-15660-f003:**
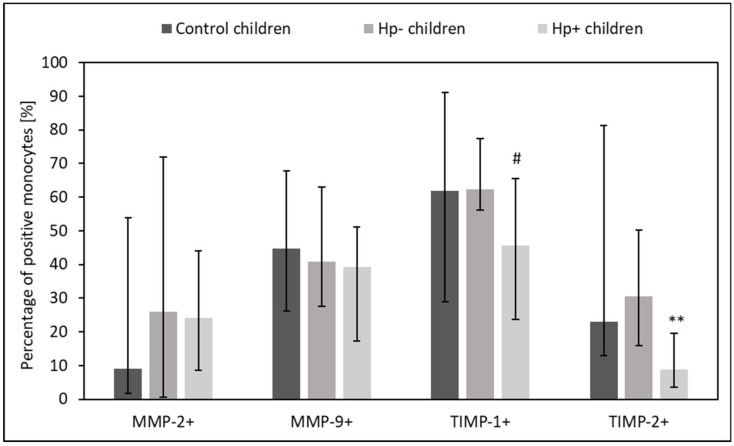
The percentage of circulating MMPs- and TIMPs- positive monocytes in Hp+, Hp-, and control groups. Hp- children: uninfected children with gastritis, Hp+ children: *H. pylori*-infected children with gastritis. Significant differences (Kruskal–Wallis test, *p* < 0.05): ** vs. Hp- children. Trend toward significance (Kruskal–Wallis test, 0.1 > *p* > 0.05): # vs. Hp- children. Variables in compared subgroups have non-Gaussian distribution, thus they are described as medians with quartiles.

**Figure 4 ijms-24-15660-f004:**
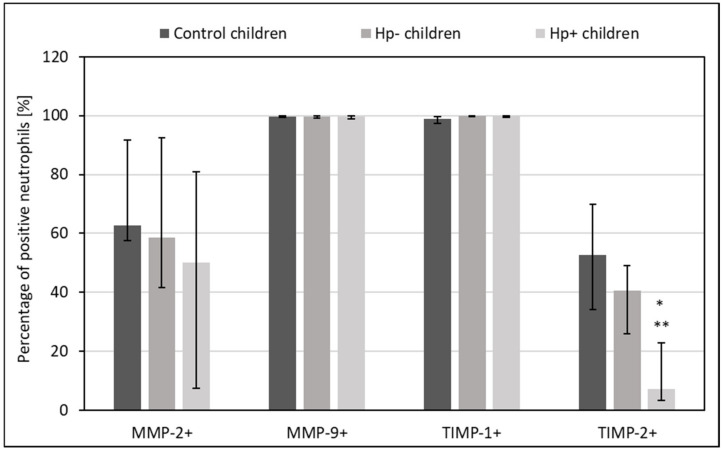
The percentage of circulating MMPs- and TIMPs- positive neutrophils in Hp+, Hp-, and control groups. Hp- children: uninfected children with gastritis, Hp+ children: *H. pylori*-infected children with gastritis. Significant differences (Kruskal–Wallis test, *p* < 0.05): * vs. control children, ** vs. Hp- children. Variables in compared subgroups have non-Gaussian distribution, thus they are described as medians with quartiles.

**Figure 5 ijms-24-15660-f005:**
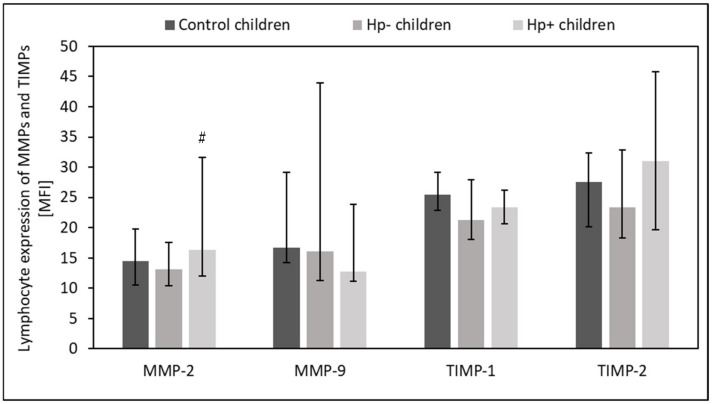
The expression of MMPs and TIMPs by circulating lymphocytes in Hp+, Hp-, and control groups. Hp- children: uninfected children with gastritis, Hp+ children: *H. pylori*-infected children with gastritis. Trend toward significance (Kruskal–Wallis test, 0.1 > *p* > 0.05): # vs Hp- children. Variables in compared subgroups have non-Gaussian distribution, thus they are described as medians with quartiles.

**Figure 6 ijms-24-15660-f006:**
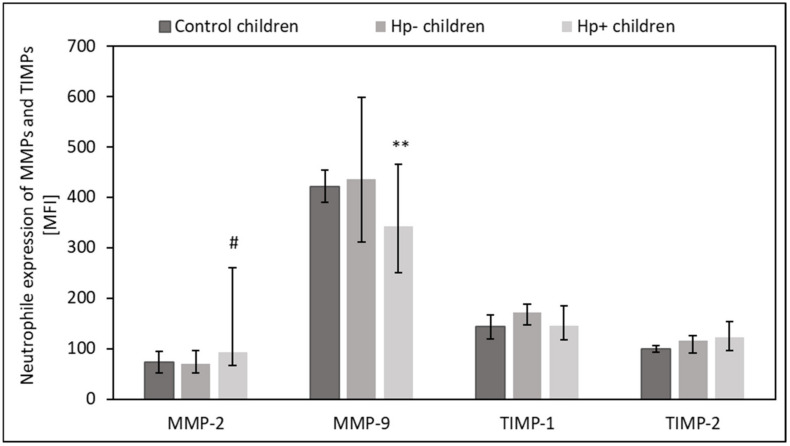
The expression of MMPs and TIMPs by circulating neutrophils in Hp+, Hp-, and control groups. Hp- children: uninfected children with gastritis, Hp+ children: *H. pylori*-infected children with gastritis. Significant differences (Kruskal–Wallis test, *p* < 0.05): ** vs. Hp- children. Trend toward significance (Kruskal–Wallis test, 0.1> *p* > 0.05): # vs. Hp- children. Variables in compared subgroups have non-Gaussian distribution, thus they are described as medians with quartiles.

**Figure 7 ijms-24-15660-f007:**
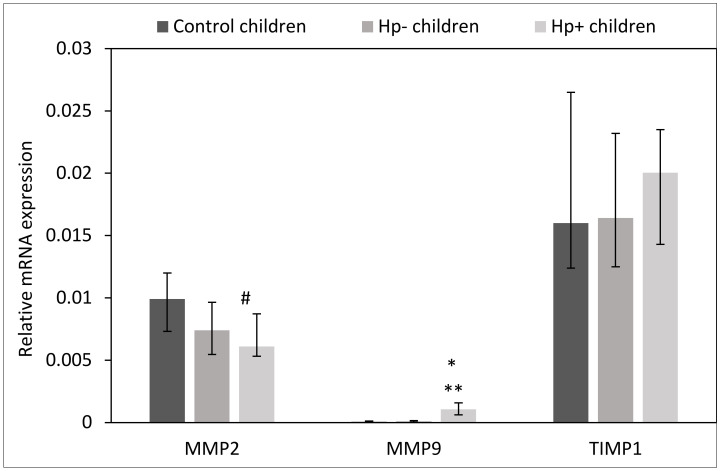
Evaluation of mRNA for MMPs and TIMP-1 in the gastric mucosa in Hp+, Hp-, and control children. Hp- children: uninfected children with gastritis, Hp+ children: *H. pylori*-infected children with gastritis. Significant differences (Kruskal–Wallis test, *p* < 0.05): * vs. control children, ** vs. Hp- children or trend toward significance (Kruskal–Wallis test, 0.1> *p* > 0.05): # vs. Hp- children. Variables in compared subgroups have non-Gaussian distribution, thus they are described as medians with quartiles.

**Figure 8 ijms-24-15660-f008:**
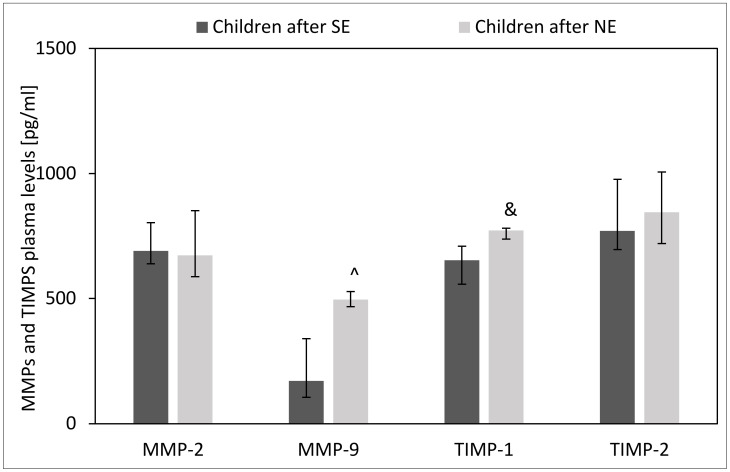
Effect of *H. pylori* eradication therapy on MMPs and TIMPs plasma levels. SE: children after successful eradication, NE: children after unsuccessful eradication. Significant differences (Mann–Whitney U test, *p* < 0.05): ^ vs. children after SE. Trend toward significance (Mann–Whitney U test, 0.1 > *p* > 0.05): & vs. children after SE. Variables in compared subgroups have non-Gaussian distribution, thus they are described as medians with quartiles.

**Figure 9 ijms-24-15660-f009:**
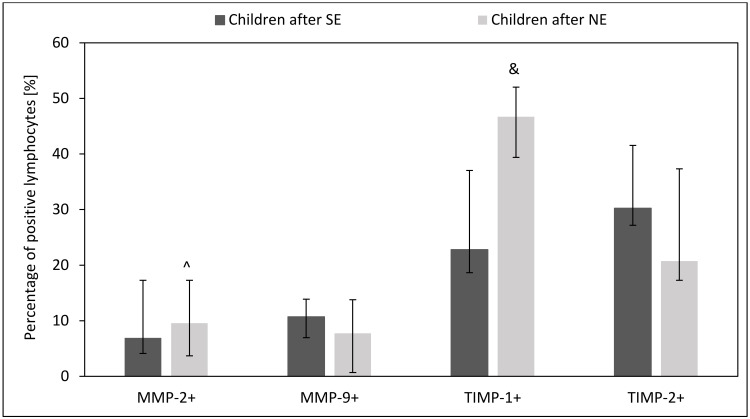
Effect of *H. pylori* eradication therapy on the percentage of circulating MMPs- and TIMPs-positive lymphocytes. SE: children after successful eradication, NE: children after unsuccessful eradication. Significant differences (Mann–Whitney U test, *p* < 0.05): ^ vs. children after SE. Trend toward significance (Mann–Whitney U test, 0.1 > *p* > 0.05): & vs. children after SE. Variables in compared subgroups have non-Gaussian distribution, thus they are described as medians with quartiles.

**Figure 10 ijms-24-15660-f010:**
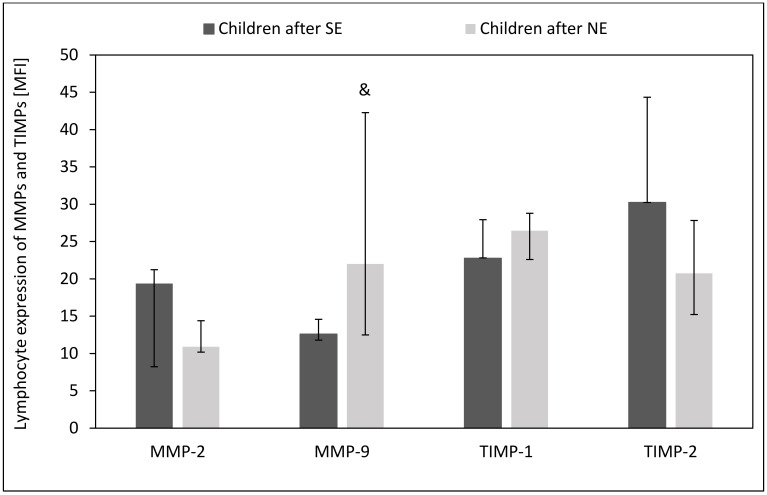
Effect of *H. pylori* eradication therapy on the intracellular expression of MMPs and TIMPs in circulating lymphocytes. SE: successful eradication. NE: unsuccessful eradication. Trend toward significance (Mann–Whitney U test, 0.1 > *p* > 0.05): & vs. children after SE. Variables in compared subgroups have non-Gaussian distribution, thus they are described as medians with quartiles.

**Figure 11 ijms-24-15660-f011:**
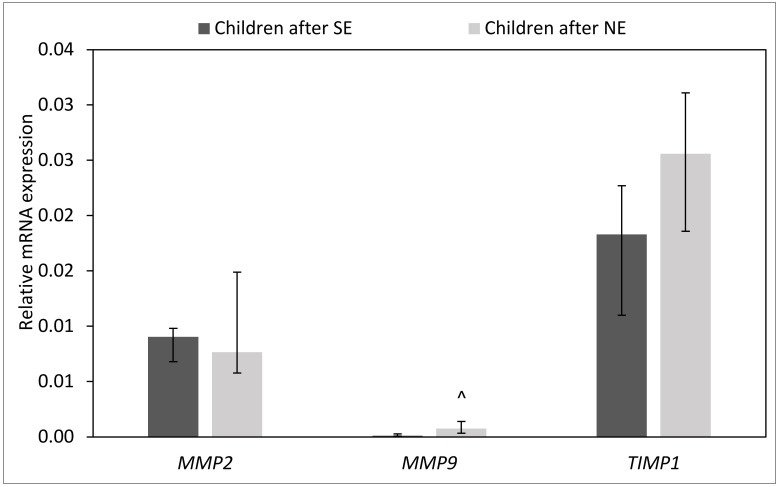
Effect of *H. pylori* eradication therapy on the expression of mRNA for MMPs and TIMP-1 in the gastric mucosa. SE: children after successful eradication, NE: children after unsuccessful eradication. Significant differences (Mann–Whitney U test, *p < 0.05*): ^ vs. children after SE. Variables in compared subgroups have non-Gaussian distribution, thus they are described as medians with quartiles.

**Figure 12 ijms-24-15660-f012:**
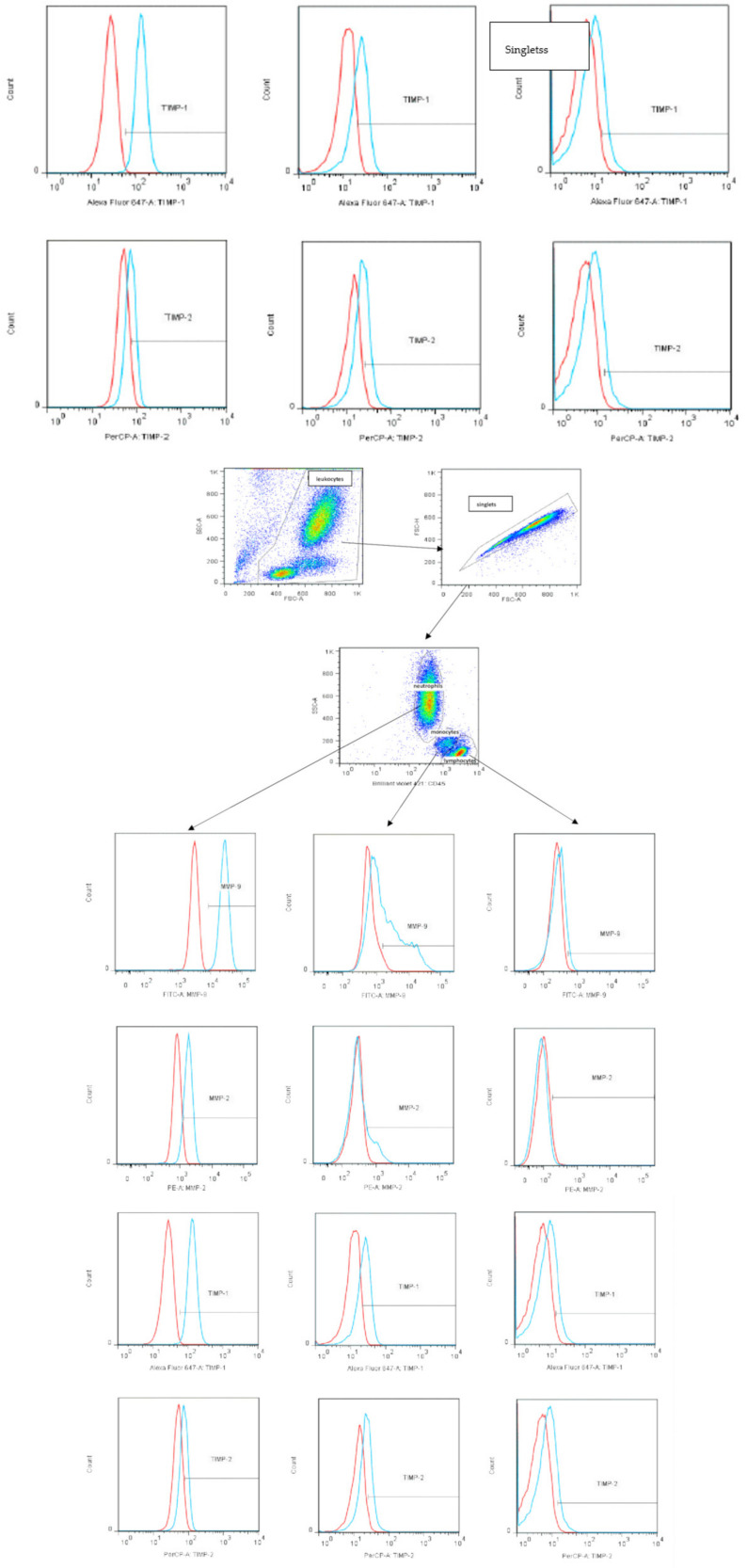
Gating strategy used to identify populations of peripheral blood leukocytes expressing MMP-2, MMP-9, TIMP-1, and TIMP-2 proteins by flow cytometry; (red lines—isotype control; blue lines—proteins expression).

**Table 1 ijms-24-15660-t001:** Spearman correlation between inflammation, activity, and density of *H. pylori* in gastric mucosa with other parameters.

Variable	Inflammation	Activity	*H. pylori*
Activity	0.19	1.00	0.02
Inflammation	1.00	0.19	0.60
*H. pylori*	0.60	0.02	1.00
Total lymphocytes [%]	0.11	0.18	0.18
MMP-2+ lymphocytes [%]	0.07	0.32	−0.04
MFI for MMP-2 in lymphocytes	−0.42	−0.23	−0.49
MMP-9+ lymphocytes [%]	0.00	0.24	−0.19
MFI for MMP-9 in lymphocytes	0.14	−0.05	0.36
TIMP-1+ lymphocytes [%]	−0.17	−0.09	0.55
MFI for TIMP-1 in lymphocytes	0.16	0.09	0.63
TIMP-2+ lymphocytes [%]	0.22	−0.20	−0.11
MFI for TIMP-2 in lymphocytes	−0.76	−0.48	−0.58
Total monocytes [%]	0.51	0.46	0.16
MMP-2+ monocytes [%]	0.00	0.15	−0.33
MFI for MMP-2 in monocytes	−0.01	−0.31	0.17
MMP-9+ monocytes [%]	0.02	−0.05	−0.22
MFI for MMP-9 in monocytes	0.22	−0.25	0.16
TIMP-1+ monocytes [%]	−0.03	0.02	0.40
MFI for TIMP-1 in monocytes	−0.02	−0.11	0.59
TIMP-2+ monocytes [%]	0.22	−0.48	0.16
MFI for TIMP-2 in monocytes	0.08	−0.45	−0.21
Total neutrophils [%]	−0.11	−0.22	−0.11
MMP-2+ neutrophils [%]	−0.02	−0.53	−0.09
MFI for MMP-2 in neutrophils	−0.38	−0.30	−0.53
MMP-9+ neutrophils [%]	0.13	0.17	0.18
MFI for MMP-9 in neutrophils	0.15	−0.59	−0.05
TIMP-1+ neutrophils %	0.06	0.00	0.32
MFI for TIMP-1 in neutrophils	−0.43	−0.05	0.06
TIMP-2+ neutrophils [%]	0.02	−0.45	−0.26
MFI for TIMP-2 in neutrophils	−0.41	−0.45	−0.78
TIMP-1 pg/mL	0.42	0.22	0.46
MMP-9 pg/mL	0.28	0.22	0.73
MMP-2 pg/mL	0.32	−0.09	0.02
TIMP-2 pg/mL	0.53	−0.22	0.09
TIMP1 mRNA	0.57	−0.04	0.32
MMP2 mRNA	0.20	−0.20	−0.05
MMP9 mRNA	0.71	−0.16	0.66

Significant correlations on the level 0.05 are marked red.

**Table 2 ijms-24-15660-t002:** Revers primers.

Gene Symbol	Gene Name	Ensembl ID/Entrez ID	Bio-Rad Unique Assay ID
*TIMP1*	Tissue metalloproteinase inhibitor 1	ENSG00000102265/7076	qHsaCID0007434
*MMP2*	Matrix metalloproteinase 2	ENSG00000087245/4313	qHsaCID0015623
*MMP9*	Matrix metallopeptidase 9	ENSG00000100985/4318	qHsaCID0011597

## Data Availability

All data generated in this article is available upon request to the first author.

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
