# Peer review of "Expression of Matrix Metalloproteinases in the Circulating Immune Cells in Children with Helicobacter pylori Infection—Correlation with Clinical Factors"

_ijms, 2023, doi:10.3390/ijms242115660_

Round 1

Reviewer 1 Report

The study provides an original approach towards the understanding of the pathogenesis of H. pylori gastritis in children. However, substantial revision is required, especially in the materials and methods section.

1. According to the MDPI author guidelines, the abstract of manuscripts sent to their journals cannot exceed 200 words. Your abstract is twice as long.

2. Please carefully check the manuscript for punctuation errors, as in various places a "." has been inserted instead of a ",".

3. Lines 75-76: "Most recent meta-analysis have shown that the overall global prevalence of H. pylori infection is 32.3%." Please provide an appropriate reference.

4. The two paragraphs in lines 83-91 do not relate to the previous paragraph, are randomly inserted within the introduction, and are not relevant for the objective of the study. Instead, more details regarding the activation of the nuclear factor kappa B-pathway should be provided within the introduction.

5. Please insert the materials and methods chapter after the introduction.

6. Ethical aspects of the study should be added as a separate sub-chapter, at the end of the "Material and Methods" section. Please add an Institutional Board ethical approval number.

7. The inclusion criteria should be detailed more. Dyspeptic symptoms are also part of functional abdominal disorders, complying to Rome IV criteria. The authors do mention that they have included children with "chronic dyspeptic symptoms suggestive of non-functional origin", but do not detail how they reached the conclusion that these symptoms were of non-functional causes.

8. Exclusion criteria mentions "previous diagnosis of other inflammatory disease, such as celiac disease, inflammatory bowel disease or allergy". How about a later diagnosis of such disorder, after inclusion in the study? How about previous PPI use?

9. Why was gastric hemorrhage an exclusion criteria, since it can be a complication of H. pylori gastritis as well?

10. Line 390, "the last one". Are the authors referring to the fifth biopsy? Please be more specific.

11. The authors mention that "The outcome of anti-H. pylori therapy was assessed at least 4 weeks after completion of therapy using 13C-UBT. " So why was an upper digestive endoscopy conducted in children  after H. pylori eradication therapy? As previously  mentioned, the inclusion criteria and the methodology of the study have not been detailed enough.

12. Please state that controls had no histhological changes of the gastric mucosa (if my assumption is accurate).

13. Please move the division of the study groups to the result section, as it was established after examination of gastric biopsies.

14. Why was there an initial antibiotic choice of metronidazole and clarithromycin? Duration of eradication scheme and ppi treatment should be mentioned as well.

15. "During endoscopy, venous blood was also obtained from each patient for immunological testing." Was the endoscopy conducted under general anesthesia? If yes, please detail among the description of the endoscopy, together with type of endoscope used and number of examiners who performed this procedure.

16. Statistical soft used for analysis should be mentioned.

17. Table 1 has not been mentioned in the manuscript's text.

18. Lines 129-132: please rewrite this sentence as it is very difficult to understand.

19. Line 292: which two studies?

20. Line 297: "gastric group"?

21. Considering the abundance of results, I believe that the conclusion chapter is too short and should be expanded.

22. English language corrections are required. Please place "H. pylori" in italics.

Moderate English language corrections are required.

Author Response

Dear Sir/Madam

We appreciate the time and effort that you dedicated to providing feedback on our manuscript and are grateful for the insightful comments on and valuable improvements to our paper. We have incorporated most of the suggestions suggested by you. Those changes are highlighted within the manuscript. Please find below a detailed point-by-point response to all comments.

Comment 1. According to the MDPI author guidelines, the abstract of manuscripts sent to their journals cannot exceed 200 words. Your abstract is twice as long.

Answer: The abstract has been significantly shortened. The headings have been removed as recommended for the authors.

Comment 2. Please carefully check the manuscript for punctuation errors, as in various places a "." has been inserted instead of a ",".

Answer: The punctuation errors were corrected.

Comment 3. Lines 75-76: "Most recent meta-analysis have shown that the overall global prevalence of H. pylori infection is 32.3%." Please provide an appropriate reference.

Answer: The appropriate reference was  provided.

Comment 4. The two paragraphs in lines 83-91 do not relate to the previous paragraph, are randomly inserted within the introduction, and are not relevant for the objective of the study. Instead, more details regarding the activation of the nuclear factor kappa B-pathway should be provided within the introduction.

Answer: The above mentioned unnecessary paragraph has been removed. More details regarding the activation of NF-kappa-B were provided.

Comment 5. Please insert the materials and methods chapter after the introduction.

Answer: The materials and methods section according to Instructions for the authors should be provided between Discussion section and Conclusions. Therefore the position of the Materials and methods chapter has been left as it was.

Comment 6. Ethical aspects of the study should be added as a separate sub-chapter, at the end of the "Material and Methods" section. Please add an Institutional Board ethical approval number.

Answer: The Institutional Board ethical approval details have been incorporated in the text as recommended by the Reviewer.

Comment 7. The inclusion criteria should be detailed more. Dyspeptic symptoms are also part of functional abdominal disorders, complying to Rome IV criteria. The authors do mention that they have included children with "chronic dyspeptic symptoms suggestive of non-functional origin", but do not detail how they reached the conclusion that these symptoms were of non-functional causes.

Answer: The inclusion and exclusion criteria and the detailed indications for upper gastrointestinal endoscopy in studied groups were correct and incorporated into the text.

Comment 8. Exclusion criteria mentions "previous diagnosis of other inflammatory disease, such as celiac disease, inflammatory bowel disease or allergy". How about a later diagnosis of such disorder, after inclusion in the study? How about previous PPI use?

Answer: The exclusion criteria were completed and added to the text. None of the patients had received antibiotics, bismuth compounds, H2 antagonists, proton pump inhibitors or immunosuppressing drugs within the last four weeks before endoscopy. Patients with previous diagnosis of coeliac disease and inflammatory bowel diseases were excluded. Only the patients who remained symptomatic after the appropriate treatment were followed. To the best of our knowledge none of them developed coeliac disease or inflammatory bowel disease.

Comment 9: Why was gastric hemorrhage an exclusion criteria, since it can be a complication of H. pylori gastritis as well?

Answer: Gastrointestinal haemorrhage was incorrectly mentioned as an exclusion criterium. None of the children included into the study had current signs of overt upper gastrointestinal bleeding, whereas two of them had a previous endoscopy because of severe upper gastrointestinal bleeding in course of peptic ulcer disease.  The text was corrected.

Comment 10. Line 390, "the last one". Are the authors referring to the fifth biopsy? Please be more specific.

Answer: “The last one” referred to the fifth gastric specimen. This was corrected to be specific.

Comment 11. The authors mention that "The outcome of anti-H. pylori therapy was assessed at least 4 weeks after completion of therapy using 13C-UBT. " So why was an upper digestive endoscopy conducted in children  after H. pylori eradication therapy? As previously mentioned, the inclusion criteria and the methodology of the study have not been detailed enough.

Answer: Only a few paediatric patients with a history of H. pylori-related diseases are considered for the control upper gastrointestinal endoscopy. In our study the second endoscopy was performed only in children with recurrence of abdominal pain (13 pt), choking as a new symptom (1 patient) and as a surveillance after upper gastrointestinal bleeding in course of peptic ulcer disease (2 patients). These data were incorporated to the text.

Comment 12. Please state that controls had no histological changes of the gastric mucosa (if my assumption is accurate).

Answer: The statement is accurate. The control group consisted of children with no signs of macroscopic and microscopic (histopathological) gastric pathology (397-8 lines).

Comment 13. Please move the division of the study groups to the result section, as it was established after examination of gastric biopsies.

Answer: The general study groups characteristics has been repositioned to the result section as recommended.

Comment: 14. Why was there an initial antibiotic choice of metronidazole and clarithromycin? Duration of eradication scheme and PPI treatment should be mentioned as well.

Answer: The schedule for the H. pylori infection treatment in children included in the study was  mistakenly stated and was corrected. All patients were treated with high dose triple therapy (amoxicillin, metronidazole and proton-pump inhibitor - PPI) for 10-14 days which is a first-choice treatment in regions where  antibiotic sensitivity is unknown or not available (as in our region) as recommended in the updated ESPGHAN/NASPGHAN guidelines (JPGN 2017). Doses of antibiotics and PPI-s were calculated based on the body-weight. In non-eradicated patients antibiotic therapy consisted of amoxicillin, clarithromycin and PPI for 10-14 days.

Comment 15. "During endoscopy, venous blood was also obtained from each patient for immunological testing." Was the endoscopy conducted under general anesthesia? If yes, please detail among the description of the endoscopy, together with type of endoscope used and number of examiners who performed this procedure.

Answer: Seventy four of 84 children, including all but one with newly diagnosed H. pylori-related gastritis, all with H. pylori negative-gastritis and all controls were examined with upper gastrointestinal endoscopy. The majority of upper  gastrointestinal endoscopies (63 of 74) were performed under general anaesthesia with flexible Olympus gastroscopes (GIF 160, GIF-H180J, GIF-Q165) by a paediatric gastroenterologist (ASP) with more than 15 years experience in paediatric endoscopic procedures. The details were incorporated into the text as recommended.

Comment 16. Statistical soft used for analysis should be mentioned

Answer: Statistical soft used for analysis was added.

Comment 17. Table 1 has not been mentioned in the manuscript's text.

Answer: Table 1 was mentioned in the manuscript in the Results section but with a different number.

Comment 18. Lines 129-132: please rewrite this sentence as it is very difficult to understand.

Answer: These lines were slightly rewritten.

Comment 19. Line 292: which two studies?

Answer: This line was rewritten.

Comment 20. Line 297: "gastric group"?

Answer: In the study (15) comparing serum levels of MMPs (including MMP-9) cited in the statement both groups of adult patients had gastritis. The correction was incorporated into the text.

Comment 21. Considering the abundance of results, I believe that the conclusion chapter is too short and should be expanded.

Answer: The conclusion chapter was expanded.

Comment 22 English language corrections are required. Please place "H. pylori" in italics.

Answer: English language correction were done. H. pylori was also rewritten in italics.

Reviewer 2 Report

1. How did the authors recruited 84 children exhibiting dyspepsia? Provide the detail. Do they outpatients? How was the symptom? Family doctors referred them?

2. Clinical manifestation was same between HP + and Hp - cases?

3. Provide the insight on the clinical significance and/insight on HP infection in children stomachs. Some people suspect early HP infection is related to early onset gastric cancer without family history. Are the authors follow up this cohort?

4. I'm not sure the prevalence of dyspepsia in Children world wide. Any geographic reason in Poland? Actually gastric cancer in East Europe looks different from those in East Asia (PMID: 36600315 ) . Provide the geographical comparison in literature on HP in children and dyspepsia in children.

Author Response

Dear Sir/Madam

We appreciate the time and effort that you dedicated to providing feedback on our manuscript and are grateful for the insightful comments on and valuable improvements to our paper. We have incorporated most of the suggestions suggested by you. Those changes are highlighted within the manuscript. Please find below a detailed point-by-point response to all comments.

Comment 1. How did the authors recruited 84 children exhibiting dyspepsia? Provide the detail. Do they outpatients? How was the symptom? Family doctors referred them?

Comment 2. Clinical manifestation was same between HP + and Hp - cases?

Answer:

All patients included to the study were hospitalized in the Department of Pediatrics, Allergology of Gastroenterology, Jurasz University Hospital in Bydgoszcz or were consulted by the paediatric gastroenterologist in Outpatient Gastroenterological Clinic for Children. All patients were recruited to the study and all endoscopies were performed by the same paediatric gastroenterologist (ASP). Patients were assigned  to the studied groups at endoscopy (when endoscopy was performed) or based on the 13C UBT result (when endoscopy was not performed - in children after eradication therapy).

Seventy four of 84 children, including all but one with newly diagnosed H. pylori-related gastritis, all with H. pylori negative-gastritis and all controls were examined with upper gastrointestinal endoscopy.

In 55 of 58 children with H. pylori -positive and negative-gastritis and in controls the main indication for upper gastrointestinal endoscopy was chronic abdominal pain and/or nausea/vomiting and at least one accompanying symptom/sign (unsatisfactory weight gain or weight loss, anaemia, abdominal pain awakening at nights, non-effectiveness of antisecretory or trimebutine treatment, positive non-invasive H. pylori testing, positive family history for coeliac disease, peptic ulcer disease or H. pylori-related gastritis). In other 3 patients the endoscopy was performed for foreign body removal (1 patient), before bariatric surgery (1 patient) and as control procedure after non-H. pylori-peptic ulcer disease (1 patients). There was no difference in clinical manifestation between children with H. pylori-positive and -negative gastritis.

In 13 of 16 children after eradication therapy in whom gastroscopy was performed, were scoped because of recurrence of abdominal pain. In other 3 patients the indication for endoscopy were choking (1 patient) and surveillance after upper gastrointestinal bleeding in course of peptic ulcer disease  (2 patients).

The majority of upper  gastrointestinal endoscopies (63 of 74) were performed under general anaesthesia with flexible Olympus gastroscopes (GIF 160, GIF-H180J, GIF-Q165) by a paediatric gastroenterologist (ASP) with more than 15 years experience in paediatric endoscopic procedures. Eleven patients underwent procedure as outpatients with only with topical aesthetic.

Comment 3. Provide the insight on the clinical significance and/insight on HP infection in children stomachs. Some people suspect early HP infection is related to early onset gastric cancer without family history. Are the authors follow up this cohort?

Answer: The clinical manifestation of H. pylori infection in paediatric population is not very clearly defined. In about 85% of infected children the bacteria remains life long asymptomatic. There is no positive association between H. pylori infection and pediatric gastrointestinal symptoms such as vomiting, diarrhoea, flatulence, chronic functional abdominal pain, halitosis, regurgitation, constipation and nausea. The evidence for an association between epigastric pain and infection has been conflicting.(6). Children with recurrent abdominal pain with any alarm symptoms independent of H. pylori status most likely have functional abdominal pain, which is one of the most commonly encountered disorders in childhood, affecting up to 25% of children worldwide. Testing for H. pylori infection in this group of patients is not recommended (7). These short characteristics of clinical manifestation of H .pylori infection in children was incorporated into the Introduction section. Only children from the cohort who remained symptomatic were followed. However, these data were not analyse.

Comment 4. I'm not sure the prevalence of dyspepsia in children world wide. Any geographic reason in Poland? Actually gastric cancer in East Europe looks different from those in East Asia (PMID: 36600315 ) . Provide the geographical comparison in literature on HP in children and dyspepsia in children.

Answer: Functional abdominal pain disorders with their four well defined entities, including functional dyspepsia are some of the most commonly encountered disorders in childhood, affecting up to 25% of children worldwide. H. pylori infection was diagnosed in 33% of Turkish children with functional dyspepsia (DOI: 10.1111/hel.12497) and in 12% of Spanish children with dyspepsia (DOI: 10.24875/GMM.M21000601). According to online survey functional dyspepsia was reported by mothers of 7.6% children recruited to the US study (J Ped 2018, 195, 134-9). To the best of our knowledge there are no current epidemiological studies focusing on the prevalence of functional dyspepsia in Poland.

Round 2

Reviewer 1 Report

The authors have addressed my comments.

Minor English language corrections are still required.